# Brief communication: Accuracy of the fallen blocks volume-frequency law

Valerio De Biagi[1]

[1]Department of Structural, Geotechnical and Building Engineering, Politecnico di Torino, Torino, Italy

*Correspondence to:* Valerio De Biagi (valerio.debiagi@polito.it)

**Abstract.** De Biagi et al. (2017) have proposed a procedure for building a fallen block volume-frequency law for rockfall phenomenon. The input data are got both from the recording of rockfall events and from the survey of fallen block volumes. The epistemic and aleatoric uncertainties present in the approach affect the value of the parameters of the law. It is shown how to quantify the errors due to missed events, to an observation period of finite duration, and to limited set of measured blocks. At the end, the procedure outputs corrective parameters for computing a design volume for rockfall analysis and engineering calculations.

## 1 Introduction

In a recent paper, De Biagi et al. (2017) have proposed a novel approach for defining a block volume-frequency law to be used in rockfall hazard quantification and in the design of rockfall protection structures. Given a "representative area" where the rockfall phenomenon occurs, the method considers the temporal occurrences of the falling block events separately from the deposit volumes distribution. The input data for implementing the procedure are got from observation and measurements. The present paper deals with the effects of epistemic uncertainties on the value of volumes predicted through the frequency law described in De Biagi et al. (2017). As summarized by Straub and Schubert (2008), the epistemic uncertainties are related to our incomplete knowledge of the process, often because of limited data. Referring to the aforementioned law, the uncertainty derives from limited number of recorded events and of surveyed blocks. In the following, the main steps required for deriving the block volume-frequency law are proposed.

1. Surveying: a the catalogue of rockfall events, $\mathcal{C}^\circ$, i.e., a catalogue containing the size of the falling blocks and the corresponding temporal information (date), and a list of fallen block measured volumes, $\mathcal{F}^\circ$, that may have fallen down at any time are drawn up. Both $\mathcal{C}^\circ$ and $\mathcal{F}^\circ$ must relate to the same representative area, i.e., the portion of deposit beyond a defined line, say the foot of the slope, where the hazard is computed.

2. Threshold volume and reduced datasets: the catalogue of the events $\mathcal{C}^\circ$ contains all the recorded events gathered in a time window of temporal length, $t^\circ$. Since there is the possibility that small events have not been always recorded, a threshold volume, $V_t$, defined as the minimum size of a fallen block that has always been observed and recorded (after its occurrence), is established. Entries related to volumes smaller than $V_t$ are removed from $\mathcal{C}^\circ$ and $\mathcal{F}^\circ$. These new datasets

are denoted as reduced catalogue, $\mathcal{C}$, and reduced list, $\mathcal{F}$, having a number of entries equal to $n$ and $N$, respectively. The temporal length $t^\circ$ is increased to consider that the observations begun with the occurrence of an event, i.e., $t = t^\circ + \frac{t^\circ}{2n}$.

3. Choice of the probabilistic models: two probabilistic models are chosen. One should be able to describe the temporal occurrences of the events of the reduced catalogue; the other the distribution of the surveyed volumes. Under the hypothesis of Poisson point process, a Poisson distribution, characterized by the parameter $\lambda > 0$, is considered for the occurrence of falling block events. A Pareto distribution is adopted for the latter, as the power laws found in the literature remark. In De Biagi et al. (2017), a generalized Pareto distribution (GPD), which can be ascribed to a Pareto type II distribution, is adopted for describing the distribution of the surveyed volumes. The two probabilistic models are merged together. That is, the following equality holds:

$$\frac{1}{\lambda T} = \bar{F}(v), \tag{1}$$

where, $T$ is the return period, $v$ is a given volume and $\bar{F}$ is the survival function of the Pareto distribution. The survival function is the complementary cumulative distribution function: $\bar{F}_V(v) = 1 - F_V(v)$.

4. Evaluation of the parameters of the distribution: the estimate of the parameters can be obtained through maximum likelihood methods from the reduced data sets. Referring to the occurrences process, the ratio $\hat{\lambda} = n/t$, i.e., the annual frequency of events larger than $V_t$, is an unbiased estimate of Poisson's occurrence parameter. Referring to the distribution of the volumes, details are provided in Section 2.

## 2   Pareto type I distribution

For dealing with a reduced number of parameters, a Pareto type I (Pareto I) distribution is adopted for describing the distribution of the surveyed volumes listed in $\mathcal{F}$. The survival function of Pareto I is

$$\bar{F}(v) = \begin{cases} 1 & v < \mu \\ \left(\frac{v}{\mu}\right)^{-\alpha} & v \geq \mu \end{cases}, \tag{2}$$

where $\mu$ is the location parameter and $\alpha$ is the shape parameter, both positive.

As reported in the literature (see, e.g., Arnold (2015) and the references cited herein), various procedures for estimating the parameters of Pareto I are proposed. In the present analysis, the threshold volume is the estimate of the location parameter, i.e., $\hat{\mu} = V_t$. This value is arbitrary and strictly depends on the possibility to observe small falling block events. Referring to the shape parameter, in Arnold (2015) , the following estimator is proposed:

$$\hat{\alpha} = N \left[ \sum_{i=1}^{N} \ln\left(\frac{v_i}{V_t}\right) \right]^{-1}, \tag{3}$$

where $v_i$ is the volume of each block measured in the representative area. Hence, the survival function becomes:

$$\bar{F}(v) = \begin{cases} 1 & v < V_t, \\ \left(\frac{v}{V_t}\right)^{-\hat{\alpha}} & v \geq V_t \end{cases},$$

(4)

Substituting Eqn. (1) into Eqn. (4), the volume $v(T)$ corresponding to a given return period (larger than $1/\hat{\lambda}$) is determined:

$$v(T) = V_t \left(\hat{\lambda} T\right)^{\frac{1}{\hat{\alpha}}}.$$

(5)

## 5  3  Reliability of the block volume-frequency law

One of the key questions that arose in the discussion of "Estimation of the return period of rockfall blocks according to their size" relates both to the minimum number of observed events and to the minimum number of surveyed blocks needed for building the curve. In the present section, the reliability of the results obtained though Eqn. (5) is discussed in the light of the consistency of the catalogue of the events $\mathcal{C}$ and of the list of measured blocks $\mathcal{F}$. In the following, the volume of the block having return period $T$ determined through Eqn. (5), with Pareto parameters vector $\boldsymbol{\pi}$, after the observation of $n$ events during $t$ time, is denoted as $\mathcal{V}(T,t,n,\boldsymbol{\pi})$.

### 3.1  Error due missed recorded events

The temporal information is relevant for establishing a link between return period and block volume. In the present section, the effects of missed events are analyzed. Supposing that $p$ events larger than $V_t$ ($p \in \mathbb{N}$) have not been observed even if they have occurred, they would be part of the reduced catalogue $\mathcal{C}$. Since the length of the observation period does not change, the value of the estimate of $\lambda$ varies: the corrected value, $\lambda_c$, is

$$\lambda_c = \frac{(2n+1)(n+p)^2}{n^2(2n+2p+1)}\hat{\lambda},$$

(6)

resulting in $\lambda_c > \hat{\lambda}$. Using the notation previously illustrated, this implies $\mathcal{V}(T,t,n+p,\boldsymbol{\pi}) > \mathcal{V}(T,t,n,\boldsymbol{\pi})$. In other words, if it is supposed that events have not been observed even if they have occurred, the volume at a given return period is underestimated.

The error due to the lack of $p$ observed events is computed in terms of ratio between the value of the volumes corresponding to the same return period, $T$, i.e.,

$$\mathcal{E}_{T,p} = \frac{\mathcal{V}(T,t,n+p,\boldsymbol{\pi})}{\mathcal{V}(T,t,n,\boldsymbol{\pi})}.$$

(7)

Substituting the known terms, the ratio of Eqn. (7) can be rewritten as

$$\mathcal{E}_{T,p} = \left[\frac{(2n+1)(n+p)^2}{n^2(2n+2p+1)}\right]^{1/\alpha} = \mathcal{E}_p.$$

(8)

It can be noted that, as expected, $\mathcal{E}_{T,p}$ is greater than one; in addition, $\mathcal{E}_{T,p}$ does not depend on the return period of the estimated volume, thus, it can be simply named as $\mathcal{E}_p$. Neither the length of the observation period enters in Eqn. (8). For practical use,

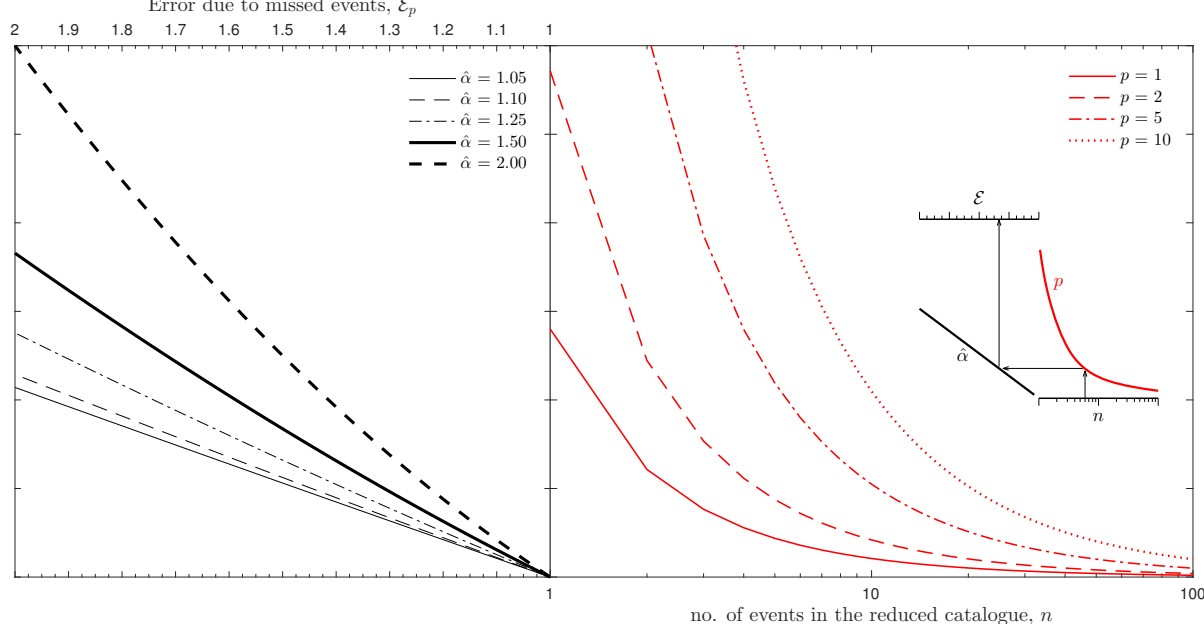

**Figure 1.** Design graph for the estimation of the error due to missed events, $\mathcal{E}_p$, as a function of the number of events $n$ in the reduced catalogue, $\mathcal{C}$, the expected number of missed events, $p$, and Pareto type I distribution shape parameter, $\hat{\alpha}$.

a design chart, which can be used in various ways depending on the required data, is proposed in Figure 1. As reported in the sketch of the plot, if the expected number of missed events $p$ and the shape parameter $\hat{\alpha}$ of Pareto I are known, the error $\mathcal{E}_p$ is derived from the number of events in the reduced catalogue, $n$ or, viceversa, given a value of the error, the consistency of the reduced catalogue, i.e., the number of events, can be determined.

### 3.2 Error due to the stochastic nature of the occurrence process

The events of the reduced catalogue $\mathcal{C}$ within the temporal range $t$ are considered to be a realization of a Poisson point process.
The error due to the aleatoric nature of the process is detailed in this section. $P_{n,t}(\pi)$ denotes the probability that, given an average annual frequency equal to $\pi$, $n$ events are observed during the period $t$, i.e,

$$P_{n,t}(\pi) = \frac{(\pi t)^n}{n!} e^{-\pi t}. \tag{9}$$

In De Biagi et al. (2017), the temporal parameter is $\hat{\lambda} = n/t$, value at which the Eqn. (9) is maximized. However, $\hat{\lambda}$ is an estimate of the parameter, which not necessarily coincides with the true value. Figure 2.A plots the value of $P_{n,t}$ against $\pi$ for different catalogues with equal $\hat{\lambda} = 0.2$. The area underlined by each curve is equal to $t^{-1}$. Figure 2.B plots the normalized curves $Q_{n,t}(\pi) = t\, P_{n,t}(\pi)$, i.e., the curves having unit underlying area.

For design purposes, given $n$ and $t$, the value $\lambda_i$, corresponding to a given $i$-percentile of the curve $Q_{n,t}(\pi)$ can be used instead of $\hat{\lambda}$. For example, considering the 90-percentile, the value of $\lambda_{90}$ to be considered is the one for which the following

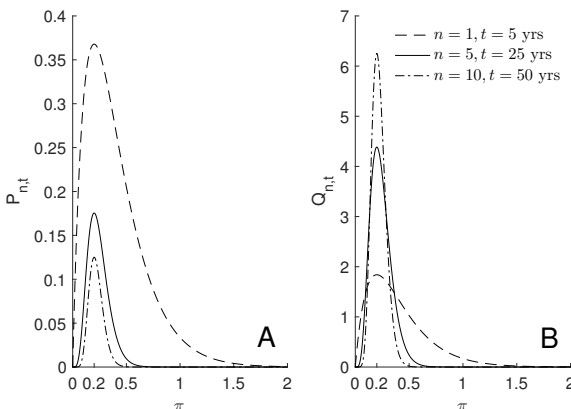

**Figure 2.** In A: Plot of $P_{n,t}$ against $\pi$; in B: plot of the normalized function, $Q_{n,t}$ against $\pi$. The legend relates to both axes.

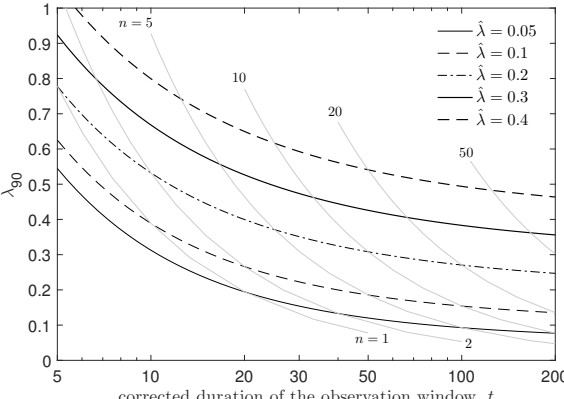

**Figure 3.** Value of $\lambda_{90}$ as a function of the duration of the observation window, $t$, or the number of events, $n$, and the estimated Poisson occurrence parameter $\hat{\lambda}$.

equalities chain subsists

$$\int_0^{\lambda_{90}} Q_{n,t}(\pi)\,d\pi = 1 - \frac{\Gamma(n+1, \lambda_{90}t)}{\Gamma(n+1)} = 0.90, \tag{10}$$

where $\Gamma(\bullet)$ is the Gamma function and $\Gamma(\bullet, \circ)$ is the upper incomplete Gamma function. In other words, if $n$ events are observed during the period $t$, there is 90% probability that the true annual frequency parameter is lower that $\lambda_{90}$. Figure 3 plots the value of $\lambda_{90}$ as a function of the duration of the observation window and the estimated $\hat{\lambda}$. It is seen that $\lambda_{90}$ tends to $\hat{\lambda}$ as much as the duration of the observation window increases.

Accounting the effects of the variability of the estimate of the frequency parameter has direct consequences on the value of the computed volume, Eqn. (5). The effects on the generated volume can be computed in terms of ratio $\mathcal{D}_{90}$ between the

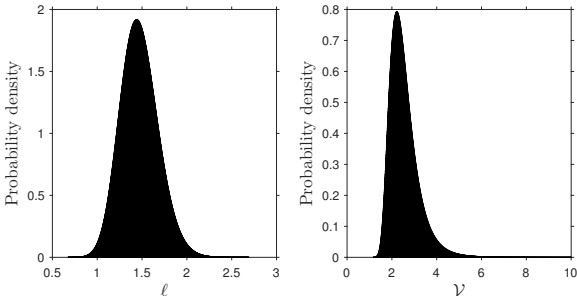

**Figure 4.** Empirical distribution of $\ell$ and the corresponding $\mathcal{U}$. The example refers to $\hat{\alpha} = 1.50$, $N = 50$, $\lambda T = 10$.

volume obtained with $\lambda_{90}$ and the reference one (i.e., with $\hat{\lambda}$),

$$\mathcal{D}_{90} = \frac{V_t \left(\lambda_{90} T\right)^{\frac{1}{\hat{\alpha}}}}{V_t \left(\hat{\lambda} T\right)^{\frac{1}{\hat{\alpha}}}} = \left(\frac{\lambda_{90}}{\hat{\lambda}}\right)^{\frac{1}{\hat{\alpha}}}. \tag{11}$$

### 3.3 Error due to a reduced number of measured blocks

5    In this section, the error related to the consistency of the list of block volumes surveyed in the representative area is determined and discussed. As already discussed in De Biagi et al. (2017), the choice of $V_t$ depends on the precision of the historical records performed in the representative area. On the contrary, the value of the estimate of the shape parameter, $\hat{\alpha}$, depends on the consistency of the reduced list $\mathcal{F}$.

   As proved by Malik (1970), the estimate $\hat{\alpha}$ follows a Gamma distribution, or equivalently, $2\alpha N / \hat{\alpha} \sim \chi^2_{2N-2}$, where $\alpha$
10  represents the real value of Pareto I shape parameter, while $\hat{\alpha}$ is its estimate. Asymptotic normality is proved since the inverse of a chi-square distribution is very close to a normal distribution.

   Considering the estimate of the shape parameter as a variate $\ell \sim \chi^2_{2N-2} \frac{\hat{\alpha}}{2N}$, fixing the values of $n$, $t$ and $T$, a distribution of volumes is expected. For example, given $\hat{\alpha} = 1.50$, $N = 50$, $\hat{\lambda} T = 10$ and $V_t = 0.5$, the volume computed through Eqn. (5) is 2.32 m$^3$. A sample of $1 \times 10^6$ values of $\ell$ were generated through a Monte-Carlo technique: left-hand side plot of Figure 4
15 shows the probability density of $\ell$ and the right-hand side axes plot the probability density of the volumes computed using the generated $\ell$. The ratio between the 90-percentile volume (3.36 m$^3$) and the reference value (2.32 m$^3$) is denoted as $\mathcal{U}_{90} = 3.36/2.32 = 1.45$.

   The error due to the consistency of the reduced list $\mathcal{F}$ is evaluated in term of ratio $\mathcal{U}$ between the generated volumes (e.g., with the $\ell$) and the reference value determined using the estimate $\hat{\alpha}$. In detail,

20   
$$\mathcal{U} = \frac{\mathcal{V}(T,t,n,\boldsymbol{\pi}^*)}{\mathcal{V}(T,t,n,\hat{\boldsymbol{\pi}})} = \frac{V_t \left(\lambda T\right)^{1/\ell}}{V_t \left(\lambda T\right)^{1/\hat{\alpha}}}, \tag{12}$$

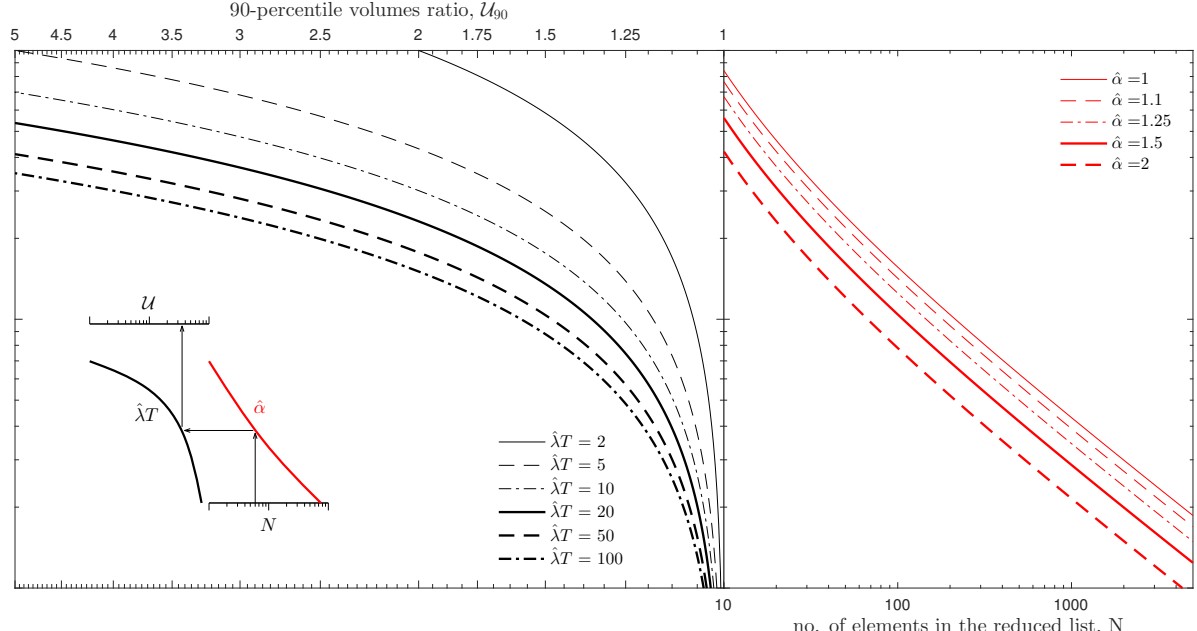

**Figure 5.** Design graph for the estimation of the volume multiplier (90-percentile) accounting for the number $N$ of measured blocks in the reduced list, $\mathcal{F}^*$, and the term $\hat{\lambda}T$, and Pareto type I distribution shape parameter, $\hat{\alpha}$.

where, $\boldsymbol{\pi}^*$ is the parameters vector containing the generated $\ell$, while $\hat{\alpha}$ is included in $\hat{\boldsymbol{\pi}}$. The previous reduces to

$$\mathcal{U} = (\lambda T)^{\frac{1}{\ell} - \frac{1}{\hat{\alpha}}} . \tag{13}$$

The value of $\mathcal{U}$ related, say, to 90-percentile can be relevant for design purposes. Because of the reciprocity of the terms composing the exponent of Eqn. (13), this corresponds to 10-percentile of the distribution of $\ell$, i.e.,

$$\mathcal{U}_{90} = (\lambda T)^{\frac{1}{\ell_{10}} - \frac{1}{\hat{\alpha}}} \tag{14}$$

where

$$\ell_{10} = \chi^2_{2N-2,0.1} \frac{\hat{\alpha}}{2N} \tag{15}$$

As for the previous case, for practical use, a design chart is proposed in Figure 5. As reported in the sketch of the plot, once the estimate $\hat{\alpha}$ is computed given $N$ measured blocks, the error corresponding to 90-percentile can be determined for different $\hat{\lambda}T$ values.

## 4   Conclusions

The quality of the survey and the precision in recording rockfall events occurring in a study area play a relevant role in computing the volumes that would probably fall by a certain period. This information is of primary importance for the design

**Table 1.** Error calculations for Buisson and Becco dell'Aquila at $T = 50$ years, supposing $p = 1$

|  | Buisson | Becco dell'Aquila |
|---|---|---|
| $n$ | 5 | 3 |
| $t$ | 25.3 yrs | 22.17 yrs |
| $\hat{\lambda}$ | 0.1976 yrs$^{-1}$ | 0.1353 yrs$^{-1}$ |
| $\hat{\alpha}$ | 0.4101 | 0.9788 |
| $v\,(50\text{yrs})$ | 40 m$^3$ | 30 m$^3$ |
| $\ell_{10}$ | 0.4173 | 0.7863 |
| $\lambda_{90}$ | 0.3366 | 0.3013 |
| $\mathcal{E}_1$ | 1.4314 | 1.39 |
| $\mathcal{U}_{90}$ | 3.7847 | 1.61 |
| $\mathcal{D}_{90}$ | 3.0695 | 2.27 |
| $V_k\,(50\text{yrs})$ | 665 m$^3$ | 665 m$^3$ |

of protection devices, for the implementation of reliability differentiation in structural engineering, for computing the risk in a certain area. From Figure 1 it emerges that, given a number of missed events $p$, the error $\mathcal{E}_p$ increases as much as the estimate of Pareto I shape parameter $\alpha$ reduces. In general, supposing that 20% of the events has not been recorded, 10 events are sufficient to keep $\mathcal{E}_{20\%} < 1.20$. Referring to the amount of measured volumes for estimating $\alpha$, the value largely depends on the return period of the expected volume, rather than to the estimate $\hat{\alpha}$. For example, having $N = 100$ and $\hat{\alpha} = 1.50$, the $\mathcal{U}_{90}$ ranges from 1.05 to 1.70 as $\hat{\lambda}T$ varies from 2 to 100. For small return periods, the consistency of the reduced list $\mathcal{F}$ can somehow be limited. Keeping $\mathcal{U}_{90}$ below 1.20 for a large range of $\hat{\lambda}T$ would imply having a reduced list containing a thousand of records.

Anyway, for design purposes, for a given catalogue of events (even limited) and a list of blocks, once the required return period $T$ is determined, the volume $V_k$ of the 90-percentile block, taking into account also the potential errors due to $p$ missed events, can be determined as

$$V_k(T) = \mathcal{E}_p\, \mathcal{U}_{90} \mathcal{D}_{90} \left[ V_t \left( \hat{\lambda}T \right)^{\frac{1}{\hat{\alpha}}} \right]. \tag{16}$$

Despite a complete error analysis for Pareto Type II is needed, referring to the examples of Buisson and Becco dell'Aquila of De Biagi et al. (2017), the error components at a return period $T = 50$ years, with a number of missed events $p = 1$, are evaluated. From Figures 4 and 5 of De Biagi et al. (2017), the volumes $v\,(50\text{ yrs})$ are approximatively $40$ m$^3$ and $30$ m$^3$, respectively. The calculations are reported in Table 1. It clearly appears that a high number of observations and surveyed blocks is fundamental for keeping the errors low.

*Acknowledgements* I kindly thank Dr Nava for the discussion on the distribution of the estimates and the two referees for their fruitful comments.

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
