# Peer review of "Brief communication: Accuracy of the fallen blocks volume-frequency law"

_Natural Hazards and Earth System Sciences, 2017_

## Referee Comment (RC1) · D. Hantz (Referee) · 14 Jun 2017

The paper deals with the important problem of the accuracy of the fallen blocks volume-frequency law, referring to a method presented in a former paper of the author (De Biagi et al., 2017). The block volumes distribution is analyzed from measurement of the size of the blocks fallen in a representative area, at any time. A Pareto distribution is used to fit the data. The error due to a reduced number of measured blocks is analyzed. The temporal occurrence of the block falls is obtained from a survey of the slope during an observation period. A Poisson law is used to describe the temporal occurrence of the block falls. The epistemic error due to missed recorded events is analyzed, but THE ALEATORIC ERROR DUE TO THE STOCHASTIC NATURE OF THE POISSON PROCESS IS IGNORED although it may be bigger than the epistemic one. For exam-

ple, in the case history of Buisson in De Biagi et al. (2017), the temporal frequency is obtained by dividing the observed number of block falls (5) by the observation period (25 years) giving a value of 0.2 block falls per year corresponding to a mean number of events (defined here as the Poisson parameter) equal to 5 for a period of 25 years. But the probability for observing 5 events with a Poisson parameter equal to 6 (0.16) is almost as high as the probability for observing 5 events with a Poisson parameter equal to 5 (0.18). So the probability for the Poisson parameter (corresponding to 25 years) to be 6 instead of 5 is far to be negligible (the same goes for Poisson parameters of 4 or 7). So a confidence interval should be determined for the Poisson parameter. In conclusion, the paper should be completed with a section analyzing the aleatoric error due to the stochastic nature of the Poisson process.

---

## Referee Comment (RC2) · Anonymous Referee #2 · 17 Jun 2017

The presented short communication deals with the error quantification of a previously proposed block volume frequency law by the same leading author (De Biagi et al., 2017). Above all the reliability of the law is explored with respect to missed events and the limited set of measured blocks.

Firstly, it is to say that the presented extension of the frequency law is an important extension. On the other hand, this short communication is for the broad readership difficult to understand as a stand-alone-publication. The readability and rigour of this contribution would highly benefit from a closer link to its predecessor publication especially to the presented data sets of Buisson and Becco dell'Aquila. As exercised out in RC1, bringing the content of the previously shown data into the here presented framework, both shows the applicability as well as helps in understanding the proposed

formalism. Here is to note, that the absence of the aleatoric error in the review of the frequency law reliability is indeed a major concern as pointed out already in RC1.

Concluding, as for scientific completeness the variability of the Poisson parameter is of greater importance. For readability sake, the presented examples should be linked more clearly to the data set presented in the preceeding publication as exercised by a certain extent already in RC1. Additionally, the abstract should conclude in a more precise way showing the link to engineering practice.

Technial corrections:

p.3, l.8: "though Eqn. (5)" should read "through Eqn. (5)"

p.3,l.20: For clarity I would propose to write "The error $\varepsilon$T,p ..."

p.4, l.1-2 & p.6, l.8 "in the sketch of the plot" should read as "... in the inset of Fig.1/3" or equivalently, due to the sketchy nature of the right hand side of the figure

p6.,l.21: Delete the "anyway", it supposes the preceding content is of minor importance.

---

## Author Comment (AC1) · 11 Jul 2017

Thank you for your insightful comments that help me to improve the quality of manuscript.

[Dr Hantz's comment] The epistemic error due to missed recorded events is analyzed, but THE ALEATORIC ERROR DUE TO THE STOCHASTIC NATURE OF THE POISSON PROCESS IS IGNORED although it may be bigger than the epistemic one. For example . . . So a confidence interval should be determined for the Poisson parameter. In conclusion, the paper should be completed with a section analyzing the aleatoric error due to the stochastic nature of the Poisson process.

[Author's response] The observation is perfectly true. Discussing the reliability of the

volume-frequency law, in particular to the Section 3.1 related to 'Missed recorded events', the effects of the epistemic error on the value of the computed volume are considered. The aleatoric error due to the stochastic nature of the Poisson process is not taken into account.

$P_{n,t}(\lambda)$ is the probability that, given an average frequency equal to $\lambda$, $n$ events are observed during the period $t$. Mathematically, $P_{n,t}(\lambda)$ can be expressed as

$$P_{n,t}(\lambda) = \frac{(\lambda t)^n}{n!} e^{-\lambda t}.$$

Fixing the number of observed falling blocks and the length of the observation period, say, for example, $n = 5$ blocks and $t = 25$ years, the probability of observing 5 events in 25 years, i.e., $P_{5,25}$, varies as much as the annual frequency changes. The maximum probability, as reported in the manuscript and in parent paper (De Biagi et al., 2017) is obtained at $\hat{\lambda} = n/t$. In Figure 1(a) the values of $P_{n,t}$ against $\lambda$ are reported for various pairs $(n, t)$, all having a ratio equal to 0.2, i.e., $\hat{\lambda} = 0.2$. As supposed, it is observed that the curves get narrow as soon as the number of observations increases (by consequence, the length of the observation period increases proportionally). The area underlined by each curve is equal to

$$\int_0^\infty \frac{(\lambda t)^n}{n!} e^{-\lambda t} d\lambda = \frac{\Gamma(n+1) - \lim_{\lambda \to \infty} \Gamma(n+1, \lambda t)}{t\, n!} = \frac{1}{t}.$$

The curves can be normalized in order to have unitary underlying area, i.e.,

$$Q_{n,t}(\lambda) = \frac{P_{n,t}(\lambda)}{1/t} =$$

Figure 1(b) shows the normalized curves. As suggested by Dr Hantz, a confidence interval should be proposed for accounting for the aleatoric uncertainty of the estimate of parameter $\hat{\lambda}$. Instead of a confidence interval, given $n$ and $t$, the value of $\lambda = \lambda_i$,

corresponding to a given $i$-percentile can be used. For example, considering the 90-percentile, the value of $\lambda_{90}$ to be considered is the one for which the following equalities chain subsists

$$\int_0^{\lambda_{90}} Q_{n,t}(\lambda)\, d\lambda = \frac{\Gamma(n+1) - \Gamma(n+1, \lambda_{90}t)}{n!} = 0.90$$

Physically speaking, if $n$ falling block events are observed during the period $t$, there is the 90% probability that the true frequency parameter is lower that $\lambda_{90}$. Figure 2 shows the normalized curves and the shaded areas corresponding to 90-percentile. It can be seen that as much as the number of recorded events increases, the $\lambda_{90}$ reduces.

Accounting the effects of the variability of the estimate of the frequency parameter has direct consequences on the value of the computed volume, cfr. Eqn. (5) of the discussion manuscript. As done for other types of uncertainty, the effects on the expected volume can be computed in terms of ratio between the volume computed using $\lambda_{90}$ and the one using $\hat{\lambda}$. This ratio, say $\mathcal{D}$, is

$$\mathcal{D} = \frac{V_t\,(\lambda_{90}T)^{\frac{1}{\hat{\alpha}}}}{V_t\left(\hat{\lambda}T\right)^{\frac{1}{\hat{\alpha}}}} = \left(\frac{\lambda_{90}}{\hat{\lambda}}\right)^{\frac{1}{\hat{\alpha}}}.$$

As suggested by Dr Hantz, the previous considerations will be inserted in an appropriate section of the manuscript.

[Figure]

**Fig. 1.** Plots of the probability of observing a given number of falling blocks in a certain amount of time. The curves in (b) are normalized insomuch that the underlying area is one.

[Figure]

**Fig. 2.** Normalized distribution curves. The shaded areas corresponding to 90-percentile.

---

## Author Comment (AC2) · 11 Jul 2017

Thank you for your insightful comments that help me to improve the quality of manuscript.

[AR2's comment] . . . The readability and rigour of this contribution would highly benefit from a closer link to its predecessor publication especially to the presented data sets of Buisson and Becco dell'Aquila. As exercised out in RC1, bringing the content of the previously shown data into the here presented framework, both shows the applicability as well as helps in understanding the proposed formalism.

[Author's response] I perfectly agree with the observation of the anonymous referee. In the revised version of the manuscript, two examples drawn from the dataset presented

in the parent paper (De Biagi et al., 2017) will be added.

[AR2's comment] Here is to note, that the absence of the aleatoric error in the review of the frequency law reliability is indeed a major concern as pointed out already in RC1.

[Author's response] Please refer to the response to Dr Hantz (RC1). As reported, the aleatoric error will be debated in a separate section in the revised manuscript.

[AR2's comment] The abstract should conclude in a more precise way showing the link to engineering practice.

[Author's response] I agree with the observation. The manuscript will be modified in the revised version of the manuscript.

---

## Author Response (AR1)

Dear Editor,

first of all I gratefully thank all the people involved in the revision of our manuscript, in particular Dr Hantz and Anonymous Referee no.2 for their suggestions and corrections to the manuscript which certainly improved its overall quality.

Following the questions of Dr Hantz and Anonymous referee no.2, a new section and a new paragraph have been added to the manuscript.

The manuscript has been updated with the observations that the two reviewers made. In the following version of the manuscript in which the relevant changes are highlighted, the modifications related to the observations and comments made by the two referees are in red. In the following "Authors' response" letter, the comments of each referee are in bold, while authors' reply is in normal character.

We hope that our revision has fulfilled editor's and reviewers' requests.

Yours truly,

Valerio De Biagi

Referee no.1 (Dr Hantz)

In the "Interactive comment" published on the online procedure on Jun 14, 2017, Dr Hantz points out the following observation:

**R1: The epistemic error due to missed recorded events is analyzed, but THE ALEATORIC ERROR DUE TO THE STOCHASTIC NATURE OF THE POISSON PROCESS IS IGNORED although it may be bigger than the epistemic one. For example…So a confidence interval should be determined for the Poisson parameter. In conclusion, the paper should be completed with a section analyzing the aleatoric error due to the stochastic nature of the Poisson process.**

AUTH: The observation is perfectly true. This aspect has been addressed in the new section 3.2.

Anonymous referee no.2

In the "interactive comment" published on the online procedure on Jun 17, 2017, anonymous referee no.2 points out few observations:

**AR2: …The readability and rigour of this contribution would highly benefit from a closer link to its predecessor publication especially to the presented data sets of Buisson and Becco dell'Aquila. As exercised out in RC1, bringing the content of the previously shown data into the here presented framework, both shows the applicability as well as helps in understanding the proposed formalism.**
AUTH: I perfectly agree with the observation of the anonymous referee. The manuscript has been revised accordingly. A reference to the datasets reported in the parent paper (i.e., De Biagi et al., 2017) has been added. The examples are reported at lines 78-82 and in Table 1.

**AR2: Here is to note, that the absence of the aleatoric error in the review of the frequency law reliability is indeed a major concern as pointed out already in RC1.**
AUTH: Please refer to the response to Referee no.1. A novel section (3.2) has been added in the revised version of the manuscript.

**AR2: The abstract should conclude in a more precise way showing the link to engineering practice.**
AUTH: The manuscript has been modified accordingly.

[revised manuscript text omitted]